# Pulmonary Vascular Complications in Hereditary Hemorrhagic Telangiectasia and the Underlying Pathophysiology

**DOI:** 10.3390/ijms22073471

**Published:** 2021-03-27

**Authors:** Sala Bofarid, Anna E. Hosman, Johannes J. Mager, Repke J. Snijder, Marco C. Post

**Affiliations:** 1Department of Cardiology, St. Antonius Hospital, 3435 CM Nieuwegein, The Netherlands; s.bofarid@students.uu.nl; 2Department of Pulmonology, St. Antonius Hospital, 3435 CM Nieuwegein, The Netherlands; a.hosman@antoniusziekenhuis.nl (A.E.H.); j.mager@antoniusziekenhuis.nl (J.J.M.); r.snijder@antoniusziekenhuis.nl (R.J.S.); 3Department of Cardiology, University Medical Center Utrecht, 3584 CM Utrecht, The Netherlands

**Keywords:** HHT, endoglin, pulmonary vascular disease

## Abstract

In this review, we discuss the role of transforming growth factor-beta (TGF-β) in the development of pulmonary vascular disease (PVD), both pulmonary arteriovenous malformations (AVM) and pulmonary hypertension (PH), in hereditary hemorrhagic telangiectasia (HHT). HHT or Rendu-Osler-Weber disease is an autosomal dominant genetic disorder with an estimated prevalence of 1 in 5000 persons and characterized by epistaxis, telangiectasia and AVMs in more than 80% of cases, HHT is caused by a mutation in the ENG gene on chromosome 9 encoding for the protein endoglin or activin receptor-like kinase 1 (ACVRL1) gene on chromosome 12 encoding for the protein ALK-1, resulting in HHT type 1 or HHT type 2, respectively. A third disease-causing mutation has been found in the SMAD-4 gene, causing a combination of HHT and juvenile polyposis coli. All three genes play a role in the TGF-β signaling pathway that is essential in angiogenesis where it plays a pivotal role in neoangiogenesis, vessel maturation and stabilization. PH is characterized by elevated mean pulmonary arterial pressure caused by a variety of different underlying pathologies. HHT carries an additional increased risk of PH because of high cardiac output as a result of anemia and shunting through hepatic AVMs, or development of pulmonary arterial hypertension due to interference of the TGF-β pathway. HHT in combination with PH is associated with a worse prognosis due to right-sided cardiac failure. The treatment of PVD in HHT includes medical or interventional therapy.

## 1. Hereditary Hemorrhagic Telangiectasia 

Hereditary hemorrhagic telangiectasia (HHT), additionally known as Rendu-Osler-Weber disease, is an autosomal-dominant inherited disease with an estimated prevalence of 1 in 5000 individuals and higher in certain regions [1]. HHT can initially present itself with spontaneous recurrent epistaxis and mucocutaneous telangiectases. However, HHT is additionally frequently complicated by arteriovenous malformations (AVMs) in the lung, brain, liver and digestive system [2]. Unfortunately, HHT is still underdiagnosed, and entire families remain unaware of available screening and treatment opportunities [2,3,4]. 

Diagnosing HHT can be done through genetic testing or by the use of the clinical Curaçao Criteria framework. The Curaçao diagnostic criteria for HHT consist of the following [5]: Frequent and recurrent epistaxis, which may be mild to severeMultiple telangiectases on characteristic sites: lips, oral cavity, fingers, and noseAVMs or telangiectases in one or more of the internal organs (lung, brain, liver, intestines, stomach, and spinal cord)A 1st-degree relative with HHT

A diagnosis of HHT is considered confirmed if at least three criteria are present, and possible with two criteria, as listed above [6].

Currently, there are five different mutations known to cause HHT; this has led to the subdivision of HHT into five subtypes [7]. These mutations do not lead to abnormal proteins but cause haploinsufficiency, which leads to a reduction in concentration of functional proteins as well as an imbalance in the TGF-β signaling pathway [8].

The various types of HHT can be subdivided based on the genetic mutation in the TGF-β signaling pathway [2]:HHT1 is caused by mutations in the ENG gene (cytogenetic location 9q34.1; OMIM187300, encoding for the protein endoglin). HHT type 1 is characterized by a higher prevalence of pulmonary and cerebral AVMs, mucocutaneous telangiectasia, and epistaxis compared to HHT type 2 [2,9,10].HHT type 2 is caused by a mutation in the ACVRL1-gene (cytogenetic location 12q13.13; OMIM600376, encoding for the ALK1 protein) and has a higher prevalence of hepatic AVMs compared to HHT type 1 [2,9,11].HHT type 3 and HHT type 4 are linked with mutations in, respectively, chromosome 5 and 7; however, the exact genes remain unknown [12,13,14].HHT type 5 is caused by a mutation in the Growth Differentiation Factor 2 gene (GDF-2) that codes for the Bone Morphogenetic Protein 9 (BMP9) (OMIM615506) which expresses an HHT-like phenotype and is therefore classified as HHT type 5 [14,15].Mutations in the SMAD4 gene (cytogenetic location 18q21.2; OMIM175050) can cause a rare syndrome that is a combination of juvenile polyposis and HHT. This mutation is only found in 1–2% of HHT patients [2,14,16].

Approximately 80% of HHT patients have mutations in the ENG and ACVRL1-gene [8,14].

Excessive TGF-β activation contributes to the development of a variety of diseases, including cancer, autoimmune disease, vascular disease, and progressive multi-organ fibrosis. The TGF-β signaling pathway is involved in many cellular processes, including cell growth, cell differentiation, apoptosis, cellular homeostasis, and others [17,18] (Figure 1). 

In endothelial cells (ECs), TGF-β can signal through two type-1 receptors: the ALK-5 pathway in which SMAD 2 and 3 are activated and the ALK-1 pathway in which SMAD 1, 5, and 8 are activated [2,7] (Figure 1). All the known mutations in genes that cause HHT are found in the TGF-β signaling pathway. 

Several studies show biphasic effect of TGF-β in ECs [19,20]. In low concentrations, TGF-β is pro-angiogenetic, while in high concentrations TGF-β inhibits angiogenesis [20,21]. ALK-1 and ALK-5 receptors play a pivotal role in the angiogenesis and quiescence of ECs [22]. Activation of the ALK-5 pathway provides a quiescent phenotype in which proliferation and migration are inhibited while stabilization of vessels by podocytes is stimulated [22]. In contrast to ALK-5, activation of the ALK-1 receptor induces proliferation and migration and increased VEGF expression [23,24,25]. ALK-1 and ALK-5 not only have opposite responses but are also co-dependent as presence of ALK-5 is necessary for maximum ALK-1 activation [22]. Studies have shown that in ALK-5 KO mice both the ALK-5 and ALK-1 pathway are defective [24,26]. Furthermore ALK-1 can directly antagonize the ALK-5/SMAD-2/3 pathway on the level of the SMADs [24,26].

Endoglin is upregulated by ALK-1 and is an accessory receptor in the TGF-β signaling pathway that is specifically expressed in the proliferation of ECs and has an opposite effect on ALK-5 [2,22,27,28]. Endoglin is necessary in the TGF-β/ALK-1 signaling and indirectly inhibits the TGF-β/ALK-5 pathway [26,29]. In addition, high concentrations of Endoglin in ECs stimulate the BMP-9 [30]. Paradoxically outside the EC, Endoglin actually inhibits TGF-β [31]. TGF-β, BMP-9 and hypoxia stimulate the increase endoglin expression in ECs [30,32].

BMP-9 is a ligand involved in the TGF-β/ALK-1 complex, high concentrations of BMP-9 in vitro and ex vivo inhibit proliferation and migration of ECs [24,33]. In vivo, however, several studies have seen that BMP-9 in combination with TGF-β induces angiogenesis [34,35]. As with TGF-β, the functional outcome of BMP-9 in ECs appears to be dependent on multiple factors including ligand concentration and cellular context [24]. 

ECs deficient in endoglin cannot mature because the balance between the ALK-1 pathway and the ALK-5 pathway is disrupted and ALK-5 predominates [28]. Since endoglin is involved in cell proliferation, migration, and maturation, vascular defects can be explained in HHT-1 by malfunctioning EC [36,37,38]. Mutations in ENG and ACVRL-1 disrupt TGF-β signaling, altering EC tubulogenesis and pericyte recruitment, causing abnormal endothelial hyperplasia and abnormal vascular morphogenesis in HHT [2,18]. 

Mutations in the ENG and ACRVL-1 genes alter the ligand–receptor interaction, creating an imbalance between Vascular Endothelial Growth Factor (VEGF). VEGF is stimulated by the ALK-5 pathway and inhibited by the ALK-1 pathway [39,40]. Although ALK-1 and ENG are expressed in angiogenesis, their expression is suppressed in adults [40]. Increase of ALK-1 signaling pathway occurs in adults in case of tumor growth, wound healing or inflammation [41,42]. 

This allows for the creation of a thin-walled arteriovenous complex that is exposed to increased arterial blood flow and increased pressure [43]. VEGF appears to play an important role in HHT-patients with a 10 times increased VEGF plasma concentration compared to non-HHT controls [14,44]. No difference was seen in VEGF plasma concentrations between HHT1 and HH2 patients [45]. Several studies with ACVRL-1-deficient mice showed that no AVMs occur if VEGF concentrations are normal [25,46]. 

Although ENG and ACRVL-1 mutations cause HHT-1 and HHT-2, it is noteworthy that HHT vascular lesions only occur in certain organs and are not expressed throughout the body [47,48]. One theory that can explain this phenomenon is the second-hit hypothesis. As with other genetic diseases, it is believed that an external trigger or second hit such as vascular damage, inflammation, infection or angiogenic stimuli must occur to cause a second genetic mutation of a healthy HHT gene, which enhances endoglin haploinsufficiency [8,49]. 

In resting EC, endoglin is present at low concentrations, but when cells are actively proliferating or during angiogenesis and embryogenesis, the endoglin concentration is increased [8,48,50,51,52]. Several studies with knock-out (KO) animal models (KO ENG and ACRVL-1) have shown that a local external trigger such as damage or stimulation of VEGF causes the formation of AVMs [8,48,53].

Thus, in the haploinsufficient HHT setting where a second hit occurs, endoglin and ALK1 do not reach the minimum concentration necessary to perform their roles in vascular damage [8,48,54,55,56]. 

Further genome analysis of HHT families with phenotype variability as well as families with HHT whose genetic causes are unknown may be useful to identify new genes that may explain the heterogenic spectrum [9]. 

## 2. Pulmonary Arteriovenous Malformations in HHT 

Pulmonary AVMs (PAVMs) are a direct connection between a pulmonary artery and pulmonary vein without the interference of the pulmonary capillary bed. This results in an intrapulmonary right-to-left shunt with no gas exchange and a reduced filtering capacity of the pulmonary capillary bed. PAVMs are frequently underdiagnosed and asymptomatic [57]. PAVMs can be present from birth and are mainly fully developed in adulthood. However, PAVMs can continue to grow later in life, for example, during pregnancy or changes in pulmonary hemodynamics [58,59]. The size of the right-to-left shunt determines the degree of hypoxemia, increased ventilation, and cardiac output (CO) [59,60,61]. PAVMs can further result in rare but severe complications like massive hemoptysis, hemothorax, cerebrovascular events, and abscesses [62]. 

The estimated prevalence of PAVMs is 38 in 100,000 individuals [63]. Approximately 15–50% of HHT patients have PAVMs [57,64,65]. However, 80–90% of the patients with PAVM have HHT as the underlying cause. The prevalence of the PAVMs in HHT depends on the type of mutation: mutations in ENG have a higher prevalence (62%) compared to mutations in ACRVL-1 (10%) [66]. PAVMs are more common in women than men, and because pregnancy is a risk factor for PAVM-related complications, it is important to screen high-risk patients for its presence. The second international clinical guideline for diagnosis and management of HHT recommended to screen all patients with possible or confirmed HHT for pulmonary AVMs [4]. Ninety percent of PAVMs have a single feeding artery and are called simple PAVMs. In 5% of cases, it concerns a complex PAVM involving two or more feeding arteries from different segments. In 5% of cases, there are diffuse PAVMs involving many feeding arteries [10,67,68].

The gold standard for screening PAVM is transthoracic contrast echography (TTCE) using agitated saline. TTCE has a sensitivity of 95–100% and can therefore be used to exclude a PAVM [4,63,69,70].

The degree of pulmonary shunt can be classified by the number of microbubbles found in the left heart. The presence of a moderate or large shunt is an independent predictor of cerebrovascular events and brain abscesses [71]. In case of a positive TTCE a chest CT pulmonary angiography should be performed to diagnose treatable PAVM [4]. However, a chest CT pulmonary angiography can be withheld in case of small right-to-left shunt because in this group either no PAVM is found or they are too small for embolization therapy [72]. 

Screening for PAVMs in (asymptomatic) HHT patients is justified due to good treatment options and non-invasive examination, reducing the risk of serious complications [4,73]. It remains unknown what the optimal screening interval is in HHT patients without a pulmonary shunt at the initial presentation [74]. The international clinical guidelines for diagnosis and management of HHT recommend treating PAVMs with transcatheter embolotherapy through the use of detachable coils or plugs, frequently preventing surgery [4]. Treatment of PAVMs is discussed further in Section 4.

## 3. Pulmonary Hypertension Caused by HHT 

PH is a condition of increased blood pressure within the pulmonary arteries. PH has been defined as a mean pulmonary arterial pressure (PAP) ≥25 mmHg at rest as assessed by right-heart catheterization [6]. In the presence of a low pulmonary artery wedge pressure (≤15 mmHg), the PH is called pre-capillary. Within the clinical classification of PH, multiple clinical conditions have been categorized into five groups as follows [6]:Group 1—Pulmonary arterial hypertension (PAH)Group 2—PH caused by left heart diseaseGroup 3—PH caused by lung diseases and hypoxiaGroup 4—Chronic thrombo-embolic PH and other pulmonary arterial obstructionsGroup 5—PH with unclear or multifactorial mechanisms

There are no data describing the prevalence of PH per group. In an echocardiographic study, the prevalence of PH (estimated pulmonary artery systolic pressure >40 mmHg) was 11%, with 79% of patients suffering from left heart disease and 10% suffering from lung diseases, respectively [6]. A study from Peacock et al. 2007 has shown that the prevalence of PAH in Europe is 15–50 subjects per million individuals in the population [6].

Diagnosing PH in HHT can be challenging. Symptoms of HHT, such as fatigue, dyspnea and exercise intolerance, resemble those of PH due to anemia, hypoxemia associated with PAVMs, inadequate sleep due to epistaxis, and the psychological burden of a chronic illness [2,6,75]. 

A transthoracic ultrasound (TTE) should always be performed when PH is suspected. TTE provides different echocardiographic variables, such as an estimation of the PAP and secondary signs, to assess the probability of PH [6]. A right heart catheterization should be performed to confirm the diagnosis if treatment of PH is being considered [6]. PH in HHT can be divided into two groups: pre-capillary PH and post-capillary PH, based on the underlying etiology. 

One of the subgroups of PAH, a disease with a pre-capillary hemodynamic profile and an increased pulmonary vascular resistance (PVR > 3 Woods Units), is heritable PAH (HPAH) [76]. Several studies have shown that the bone morphogenetic protein receptor type II (BMPR2) gene is mutated in 70% of cases with HPAH [77,78,79,80]. The BMPR2 is a receptor of the TGF-β superfamily because it seems to be a ligand which influences cytokines involved in proliferation, migration, differentiation, and apoptosis [76]. ECs additionally regulate vascular function by controlling the production of vasoconstrictors, vasodilators, and the activation and inhibition of smooth muscle cells (SMCs) [2]. BMPR2 is a serine/threonine receptor kinase involved in EC apoptosis and prevents arterial damage and unfavorable inflammation [15,81,82,83]. Dysfunction of the BMPR2 gene results in hypertrophy of the SMC, deposition of the extracellular matrix, the proliferation of endothelial cells, and an increase in adventitial fibroblasts. This leads to endothelial dysfunction, a decreased production of vasodilator and anti-proliferative agents (NO and prostaglandin), and an increase of the production of agents that promote vasoconstriction and proliferation (thromboxane A2 and endoteline-1). This results in an increased PVR and leads to right ventricular overload, hypertrophy, and dilatation, eventually leading to death [2,6]. 

Less than 1% of patients with HHT suffered HPAH caused by a mutation in the ACVRL1 gene [9]. Up to 20% of the known ACVRL1 mutations are associated with the development of PAH. 

Research by Vorselaars et al. showed that very few cases are known in the literature with the combination PAH-HHT. Ref. [2] in total, 113 cases have been described in the literature of which 18 patients have an ENG mutation (PAH-HHT1) and 79 patients an ACVRL1 mutation (PAH-HHT2). No data on the prognosis are described in literature on patients with PAH-HHT1, but it is expected that the prognosis is worse compared to patients with only HHT. Patients with PAH-HHT2 often present younger and have a worse prognosis. For example, 28% of patients with PAH-HHT2 are <18 years old [2].

However, a majority of family members of patients with HPAH in combination with HHT do not develop HPAH, indicating that other genetic or environmental factors are required to develop an HPAH phenotype [84].

Symptomatic HPAH patients with ACVRL1 mutations, frequently without HHT, are more likely to present with symptoms than patients with a BMPR2 mutation or idiopathic PAH [85]. Despite the fact that patients with HPAH respond more ideally to treatment with vasodilators, HPAH develops more progressively compared to patients with idiopathic PAH [79,86]. A study by Lee et al. has shown that patients with PAH-HHT have significantly lower three-year survival rates compared to patients with a BMPR2 mutation or idiopathic PAH (53% vs. 74%) [87].

Post-capillary PH arises from a hyperdynamic state caused by an increased cardiac output (CO), which can cause heart failure in the long term [6]. Within HHT, the increased CO (sometimes up to three times normal) is frequently caused by the HAVM-related shunt [75,88]. HAVMs occur in 32–78% of patients with HHT and are mainly seen in HHT2 caused by ACVRL1 mutation [2,71]. In addition to HAVM, the CO can also be increased due to anemia caused by epistaxis and gastrointestinal bleeding, which is frequent in patients with HHT. International guidelines recommended that screening for HAVM should be offered to adults with definite or suspected HHT. Doppler ultrasonography, multiphase contrast CT and MRI can be used to screen HAVM [4].

## 4. Treatment and Follow-Up of Pulmonary Vascular Disease in HHT

Treatment of PAVMs in HHT is recommended to prevent severe complications—in particular, the development of brain abscesses and cerebral ischemic events—and is therefore justified for asymptomatic patients. In addition, symptoms of hypoxia and dyspnea can be reduced by PAVM treatment [4]. Embolization therapy is done by the transcatheter vaso-oclusion of PAVMs through the use of detachable coils or plugs. Complications that can arise with vaso-occlusion are pleural pain and pleural effusion, which improve when treated symptomatically [89]. Because a right–left shunt increases the risk of infections and ischemic events, treatment of PAVM is indicated when the supply vessels are ≥3 mm or as small as technically feasible. Cannulation and embolization of smaller vessels can be more challenging. It is important to embolize as distal as possible to ensure that other well-functioning branches of the pulmonary vasculature are not occluded as well [90,91]. In the long term (2–21 years), the vaso-occlusion success rate is 75%. This is probably due to development of other supply vessels that were previously not visible during the initial procedure or growth of other PAVMs [92]. International guidelines recommended providing long-term follow-up in patients with PAVMs, in order to detect growth of untreated PAVMs. It also recommended to advise patients to use antibiotics before procedures with a risk of bacteremia, avoid SCUBA diving and taking extra care when intraveneous access is in place to avoid air emboli [4].

Sometimes, the PAVM are complex and diffuse involving pulmonary arteries from different segments. This group is difficult to treat; surgery might be an alternative for percutaneous treatment. However, lung transplantation might be the only option left [90], [91].

Despite the fact that local treatment of telangiectases and PAVMs continues to improve, no ideal systemic therapy is available to date. Various studies and trials have attempted to find new drugs and have investigated the possibilities for repurposing existing drugs [2,93].

Currently, anti-angiogenic drugs used in cancer treatment (anti-VEGF antibodies and thyrosine kinase inhibitors) are under investigation with the aim of inhibiting the pro-angiogenic processes in HHT (Figure 2).

VEGF plays a role in the development of AVMs and anti-VEGF therapy has shown to be effective in the treatment of other AVMs. Several case reports described treatment of diffuse PAVM with bevacizumab in which respiratory symptoms improved and epistaxis was decreased, without the formation of new AVMs on chest-CT during follow-up [94]. Bevacizumab is a monoclonal antibody used in the treatment of cancer. This antibody acts on endothelial growth factor (VEGF) to inhibit neoangiogenesis. VEGF promotes angiogenesis and interacts with ECs. In case of tumors, wound healing and HHT high VEGF concentrations are observed [14,44,95,96,97]. Although these results are hopeful, further scientific research is needed to use bevacizumab in the treatment of PAVMs. 

Treatment differs for the different types of PH in HHT. There have not yet been randomized control trials to contribute to a guideline for PAH-specific therapy in HHT [2]. Standard therapy of PAH is currently recommended in patients with heritable PAH based on HHT specific mutations. The aim of current treatment is to unload the right ventricle and reduce symptoms, thus improving quality of life. 

Treatment of PAH consists of lifestyle advice and drug therapy. Lifestyle advice includes avoiding pregnancy and infections, having elective surgery performed in specialized centers with experience in PH, genetic testing of family members, oxygen, psychological assistance, and water and salt reduction [4,6]. Drug therapy for PAH consists of calcium channel blockers for patients responding well to invasive vasodilation test, endothelin receptor antagonists, phosphodiesterase type 5 inhibitors, soluble guanylate cyclase stimulators, and prostacyclin [2,6]. In young therapy-resistant patients, lung transplantation can be performed as well. 

Different drugs are currently under investigation for the treatment of PAH. Tacrolimus is a drug used to prevent rejection after allogenic organ transplantation [98]. The precise effect of tacrolimus is unknown. It is thought that tacrolimus engages in the BMP9-ALK1-ENG-SMAD pathway and stimulates the transcription of ENG and ALK-1 in ECs, reducing haploinsufficiency [8] (Figure 2). In addition, it appears that tacrolimus activated the BMPR-2 signalling pathway which is suppressed in PAH [98,99]. Research by Albinana et al. showed that after treatment with tacrolimus the ECs had an increase in mRNA expression as well as the protein endoglin and ALK-1. This stimulates the TGF-β /ALK-1 pathway and ECs functions such as tubulogenesis and cell migration [97,100,101]. Furthermore reduced VEGF activity was also seen in animal models [97,102].

A few case reports describe the use of tacrolimus in PAH with promising results [98]. A phase 2b randomized control study by Spiekerkoetter et al. 2017 showed an increase of BMPR2 expression, improvement of the 6-min walk distance, serological and ultrasound parameters of heart failure. However, these improvements were not significant probably due to small group size and requires further research [103].

A recent case report has shown that low-dose tacrolimus treatment improved HHT-related epistaxis but had no effect on PH progression in HHT patients [104]. However, tacrolimus is poorly soluble in plasma and has a low bioavailability, so local availability in the lung may be not optimal since it is given in a low dose [105]. Systemic therapy with tacrolimus can also cause side effects of neuro- and nephrotoxicity. A study by Wang et al. showed that it is possible to administer tacrolimus in aerosol form by using biodegradable polymeric acetalated dextran nano particles (Ac-Dex NP), which can deliver tacrolimus deep into the lungs [106]. Due to local delivery and therefore bypassing the first pass effect, local concentrations of tacrolimus might be achieved with less risk of systemic side effects. 

PH due to left heart disease can be treated by means of salt reduction and diuretics. Embolization of liver AVMs can cause serious complications, such as biliary ischemia. Treatment of choice might be the use of intravenous bevacizumab, recently recommended in the international guideline for HHT [4]. Several small non-randomized studies show that the use of bevacizumab in PH due to hepatic AVMs improved the cardiac output, pulmonary arterial pressure, left ventricular filling pressure and reduced the progression of HAVMs [4,107,108,109]. 

Tacrolimus has demonstrated to be a potent ALK-1 signaling mimetic, downregulating the ALK-1 loss of function transcription response, tacrolimus is therefore an interesting option for the treatment of HAVM in HHT2 with high cardiac output PH [8]. 

Secondly, if anemia is present the underlying etiology should be treated to reduce the high CO.

## 5. Conclusions

In this review, we discussed the pathophysiology, screening, and treatment of PVD, both PAVM and PH, in HHT. Research into pathophysiology of these mutations has led to potential targets for therapy like tacrolimus and bevacizumab. Although case reports show promising results, scientific evidence is still insufficient to use these therapies in daily practice. Further research is required, and it is reasonable to assume that clinical trials will follow.

## Figures and Tables

**Figure 1 ijms-22-03471-f001:**
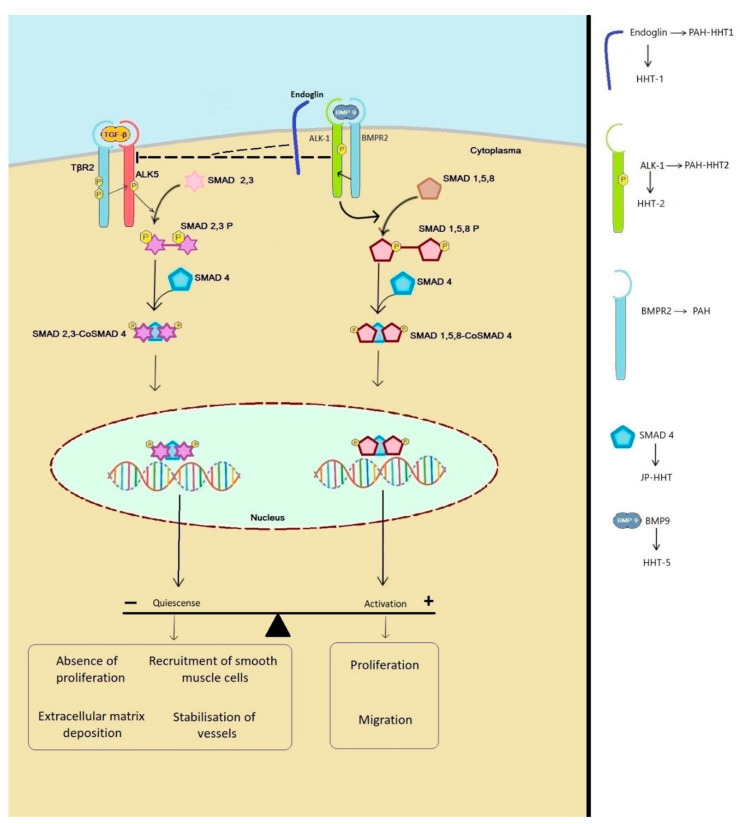
Molecular pathophysiology of hereditary hemorrhagic telangiectasia (HHT). Physiological signaling in endothelial cells occurs through ALK-1 and ALK-5. Signaling through ALK-1 results in the activation of the SMAD 1,5,8-CoSMAD4 pathway resulting in proliferation and migration of endothelial cells (ECs). ALK-1 (and indirectly endoglin) also inhibits ALK-5 signaling. ALK-5 signaling activates the SMAD 2,3-CoSMAD4 pathway which inhibits proliferation and migration of ECs and stimulate the stabilization of the vessels.

**Figure 2 ijms-22-03471-f002:**
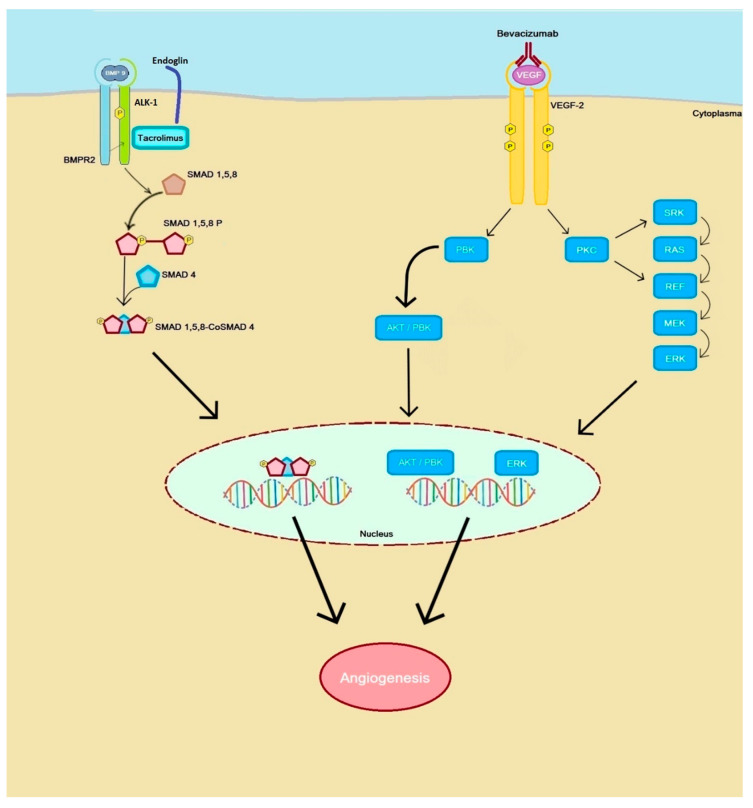
Schematic model of the targets of bevacizumab and tacrolimus. Bevacizumab binds VEGF signaling molecules that prevent (vascular endothelial growth factor) VEGF from binding effectively to the VEGFR2 receptor and as a result it reduces neoangiogenesis. Bevacizumab enhances the BMP9-ALK1-ENG-SMAD pathway and enhances ENG and ALK1 expression.

## Data Availability

No new data were created or analyzed in this study. Data sharing is not applicable to this article.

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
