# Peer review of "Pulmonary Vascular Complications in Hereditary Hemorrhagic Telangiectasia and the Underlying Pathophysiology"

_ijms, 2021, doi:10.3390/ijms22073471_

Round 1

Reviewer 1 Report

This manuscript described about current molecular pathology and treatment pf Hereditary Hemorrhagic Telangiectasia(HHT). Interestingly this manuscript explained about HHT2 with pulmonary arterial hypertension(PAH) and its treatment now and in future(tacrolims, bevacizumab). The manuscript was easy and simple to reading. 

But there were some minor spelling miss and, I think the authors ignore HHT1 with PAH.

In Figure 1 and 3, term 'ALK-1' were mistyped as 'AKL1'

In page 7, the authors described about PAH with HHT caused from AVCRL1 mutation, but in limited cased the mutations of ENG also cause PAH with HHT and in the guideline about  classification of PAH(6th world symposium of pulmonary hypertension in Nice, 2018), heritable PAH with ENG mutation was clearly included.

 The authors did not mention that ENG mutation also cause PAH. I think PAH with ENG mutation also should be described.

Author Response

  1. In Figure 1 and 3, term 'ALK-1' were mistyped as 'AKL1

Thank you, we have corrected the mistake.

  1. In page 7, the authors described about PAH with HHT caused from AVCRL1 mutation, but in limited cases the mutations of ENG also cause PAH with HHT and in the guideline about  classification of PAH(6th world symposium of pulmonary hypertension in Nice, 2018), heritable PAH with ENG mutation was clearly included.

Thank you, we have added text to include ENG mutations and HPAH.

  1. The authors did not mention that ENG mutation also cause PAH. I think PAH with ENG mutation also should be described.

Thank you, we have included this in the manuscript.

Reviewer 2 Report

In this Review, the authors focus on the HHT and lung. They summarize how the TGFb pathway is deregulated in HHT disease and how the HHT can cause different types of pulmonary vascular diseases with the formation of AVMs as well as Pulmonary Hypertension. The authors are more focused in explaining the clinical techniques and treatments for these patients as well as methods to follow up these patients. 

The first part of this review (Signaling pathway and AVMs, points 1 and 2) needs to be improved. The second part (From point 3) is well taken, focused in a more clinical aspect and well organize.

My suggestions to improve this manuscript, are the following major and minor comments:

  1. Authors should reconsidered the title to add some clinical aspects. Although they explain the signaling pathway in a general way they are more focused on the diseases in lung, clinical, follow up, and treatment description.
  2. Fig1 needs some modification and clarification. As Eng is a Type III receptor which modulates both signaling pathways, please draw this in the Figure.  The explanation for the figure has to be improved too. There are several references that support ALK1 signaling in angiogenesis that can be used to add in the text too.
  3. Page 3. It is not completely true that ALK1 is involved in angiogenesis and ALK5 in vasculogenesis. Both processes share similarities and it is known that ALK1 is more related with proliferation, angiogenesis and migration and contrary, ALK5 is more related with vessel stabilization and maturation. I would recommend explaining better this dichotomy. Therefore, authors should better support their claim of angiogenesis and mention the controversy with some other groups, and the fact that BMP9 through ALK1 and ENG induces quiescence too. The final processes regulated in the figure must be clarified in the text with references.
  4. Page 3. ALK5 has always been related to vascular quiescence, and extracellular matrix deposition, not vasculogenesis. I understand that vasculogenesis will be in the embryonic state and if this is the case, it must be well clarified.
  5. Page 4. Clarify if possible with some reference that TGFb is not considered always an antiangiogenic factor. Depends of the receptor-interaction as well as levels of TGFb.
  6. Page 9-10. Regarding Bevacizumab, Tacrolimus and the use of other drugs, there are some HHT-treatment general reviews from 2020 that could be good to include as references and can help to improve this explanation of treatments for HHT and its action mode in the signaling pathways.
  7. The text is well followed and well-structured but it is important to revise some grammatical errors:      -page 4. Second paragraph. Second line: change Vasculair to Vascular. In that same paragraph, line 6 please remove the n from an, in the sentence that begins "with an 10 times increase VEGF". Along the same lines, it is necessary to separate "concentration plasma" that are united.      -Page 4 in paragraph 3 the last two lines are wrong. Please correct that, precisely the reference that authors give is the only one in which it has been found that in some telangiectasias there is a double mutation in humans.      -Page 6, first paragraph, at the end of line 6, a "d" is missing in organize, it would be organized.      -Page 6 paragraph that begins with "One of the subgroups...", in line 6 it is written "The BMPR2 gene belongs to the TGFbeta superfamily...". Please, clarify that it is a receptor of the TGFbeta superfamily, because it seems to be a ligand, as it is written.        -Page 8, first paragraph, line 18, where it says "during the initial procedure or grow...", it should be "growth".    -Page 9, first paragraph, at the end of line 7, remove the "s" from factors. VEGF is an endothelial growth factor. 

Author Response

  1. Authors should reconsidered the title to add some clinical aspects. Although they explain the signaling pathway in a general way they are more focused on the diseases in lung, clinical, follow up, and treatment description.

Thank you for your suggestion; we have changed the title to fit the general message of the article better.

  1. Fig1 needs some modification and clarification. As Eng is a Type III receptor which modulates both signaling pathways, please draw this in the Figure.  The explanation for the figure has to be improved too. There are several references that support ALK1 signaling in angiogenesis that can be used to add in the text too.

We have altered the figure and elaborated on the molecular pathway in the figure legends to clarify the figure better.

  1. Page 3. It is not completely true that ALK1 is involved in angiogenesis and ALK5 in vasculogenesis. Both processes share similarities and it is known that ALK1 is more related with proliferation, angiogenesis and migration and contrary, ALK5 is more related with vessel stabilization and maturation. I would recommend explaining better this dichotomy. Therefore, authors should better support their claim of angiogenesis and mention the controversy with some other groups, and the fact that BMP9 through ALK1 and ENG induces quiescence too. The final processes regulated in the figure must be clarified in the text with references.

Thank you for your comment. We have rewritten the text to elaborate and add more nuances in the role of ALK-1 and ALK-5.

  1. Page 3. ALK5 has always been related to vascular quiescence, and extracellular matrix deposition, not vasculogenesis. I understand that vasculogenesis will be in the embryonic state and if this is the case, it must be well clarified.

Thank you for your comment, we have corrected and clarified the text.

  1. Page 4. Clarify if possible with some reference that TGFb is not considered always an antiangiogenic factor. Depends of the receptor-interaction as well as levels of TGFb.

Thank you for your comment. We have clarified the role of TGF- β in angiogenesis.

  1. Page 9-10. Regarding Bevacizumab, Tacrolimus and the use of other drugs, there are some HHT-treatment general reviews from 2020 that could be good to include as references and can help to improve this explanation of treatments for HHT and its action mode in the signaling pathways.

Thank you for your comment. We have rewritten the text to elaborate the explanation Bevacizumab and Tacrolimus in HHT and included references.

  1. The text is well followed and well-structured but it is important to revise some grammatical errors:      -page 4. Second paragraph. Second line: change Vasculair to Vascular. In that same paragraph, line 6 please remove the n from an, in the sentence that begins "with an 10 times increase VEGF". Along the same lines, it is necessary to separate "concentration plasma" that are united.      -Page 4 in paragraph 3 the last two lines are wrong. Please correct that, precisely the reference that authors give is the only one in which it has been found that in some telangiectasias there is a double mutation in humans.      -Page 6, first paragraph, at the end of line 6, a "d" is missing in organize, it would be organized.      -Page 6 paragraph that begins with "One of the subgroups...", in line 6 it is written "The BMPR2 gene belongs to the TGFbeta superfamily...". Please, clarify that it is a receptor of the TGFbeta superfamily, because it seems to be a ligand, as it is written.        -Page 8, first paragraph, line 18, where it says "during the initial procedure or grow...", it should be "growth".    -Page 9, first paragraph, at the end of line 7, remove the "s" from factors. VEGF is an endothelial growth factor. 

Thank you for attentiveness, we have changed all misspellings and grammatical errors.
